# Sorafenib/2800Z Co-Loaded into Cholesterol and PEG Grafted Polylysine NPs for Liver Cancer Treatment

**DOI:** 10.3390/ph16010119

**Published:** 2023-01-13

**Authors:** Chen Zhang, Wu Zhong, Ying Cao, Bohao Liu, Xiaojun Tao, Zhuan Li

**Affiliations:** 1The Key Laboratory of Study and Discovery of Small Targeted Molecules of Hunan Province, Hunan Normal University School of Medicine, Changsha 410013, China; 2The Key Laboratory of Model Animals and Stem Cell Biology in Hunan Province, Hunan Normal University School of Medicine, Changsha 410013, China

**Keywords:** polylysine, sorafenib, SIRT7 inhibitor, GSH-sensitive PEG, tumor microenvironment, targeted therapy

## Abstract

The treatment of liver cancer remains challenging due to the low responsiveness of advanced cancer to therapeutic options. Sorafenib is the first line chemotherapeutic drug for advanced liver cancer but is frequently associated with severe side effects lead to discontinuation of chemotherapy. We previously developed a specific SIRT7 inhibitor 2800Z, which suppressed tumor growth and enhanced the chemosensitivity of sorafenib. In this study, we constructed polylysine polymer nanoparticles modified with cholesterol and GSH-sensitive PEG (mPssPC) to load sorafenib (SOR) and the SIRT7 inhibitor 2800Z to form dual-loaded NPs (S2@PsPCs) to reduce the toxicity and increase efficacy of sorafenib in liver cancer. The average size of S2@PsPC NPs was approximately 370 nm and the zeta potential was approximately 50–53 mV. We found that the release of the drugs exhibited pH sensitivity and was significantly accelerated in an acid release medium simulating the tumor environment. In addition, S2@PsPC NPs inhibited the proliferation and induced apoptosis of liver cancer cells in vitro. An in vivo study further revealed that S2@PsPCs showed high specificity to the liver cancer but low affinity and toxicity to the main organs including the heart, kidneys, lungs, and liver. Our data thus further approved the combination of a SIRT7 inhibitor and sorafenib for the treatment of liver cancer and provided new drug delivery system for targeted therapy.

## 1. Introduction

During recent decades, great efforts have been made to improve the treatment of liver cancer [1,2]. However, liver cancer therapy remains challenging due to the low responsiveness of advanced cancer to therapeutic options [3]. Sorafenib (SOR) is the first approved tyrosine kinase inhibitor (TKI) for treatment of terminal stage hepatocellular carcinoma and metastatic renal cell cancer and still the first-line chemotherapeutic drug for liver cancer [4]. Sorafenib is a multi-kinase inhibitor with activities against vascular endothelial growth factor receptor (VEGF), platelet-derived growth factor receptors (PDGFR), tyrosine kinase signaling pathways, and rapid accelerated fibrosarcoma (RAF) kinase [5]. Oral administration of sorafenib significantly improves life quality and overall survival of patients with terminal stage liver cancer [6]. However, due to the broad inhibition of kinases that are critical for physiological function and homeostasis in many organs, sorafenib administration is frequently associated with variety of adverse effect including diarrhea, hypertension, and hemorrhage, which can lead to life-threatening situations as well as discontinuation of chemotherapy [7].

SIRT7 is a NAD^+^-dependent class III histone deacetylase (HDAC III), which is predominantly localized in the nucleus where it acts as a histone deacetylase and regulates RNA polymerase I transcription [8]. Emerging evidence also implicates SIRT7 in cancer biology by inducing H3K18 deacetylation for maintaining fundamental properties of the cancer cell phenotype [9,10,11]. Elevated SIRT7 levels are frequently observed in human cancers, which correlate with poor clinical outcomes and survival [12,13,14]. In liver cancer, SIRT7 expression is also upregulated in a large cohort of HCC patients [10], and we have identified that elevated SIRT7 expression is associated with chemosensitivity by regulating TP53 activity [15]. By using structure based virtual screening, we developed a specific SIRT7 inhibitor 2800Z, which enhanced the chemosensitivity of sorafenib and reduced the tumor burden [16], indicating that a combination of sorafenib and 2800Z could serve as new therapeutic option in the treatment of human liver cancer.

Co-delivery systems by nanotechnology are frequently used to reduce toxicity and increase the efficacy of chemotherapeutic drugs [17]. Poly-lysine (PLLS) is an endogenous amino acid and has many properties including natural biological affinity, tissue compatibility, and easy modification. It has been increasingly used as a nanomaterial with a high surface charge and long-term stability [18]. The surface positive charge of PLLS nanoparticles (NPs) facilitates cell membrane absorbance and high efficiency cellular uptake [19,20]. Meanwhile, the cationic nanomaterials can increase protons in the endosomes to facilitate acid sensitive drug release and endosome escape [21,22]. Of note, unmodified NPs are frequently adsorbed by complement or captured by immune cells, which decreases the efficiency of drug delivery [23,24]. To solve such problems, PEG-modification is often used to form a hydration shell on NPs surface, which can prolong the blood circulation time [25,26] and to ensure that NPs have enhanced permeability and retention (EPR) effects in tumor tissue [27].

Conventional drug delivery systems for targeted tumor therapy are facing challenges such as low efficiency, poor tumoral penetration, and uncontrolled drug release, mainly due to the tumor microenvironment (TME), including non-satisfactory blood circulation and low pH [28]. Intelligent nano delivery system such as TME responsive NPs have aroused great interest in approving targeted drug delivery systems [28]. By introducing various modifications, NPs can respond to different TMEs for more efficient tumor targeting [29,30,31]. Among those modifications, the GSH responsive disulfide bond is one of the most widely used nano components in intelligent nano delivery systems [32], which can be used to graft PEG and PLLS to form NPs. Most importantly, PEG will be taken off in response to a high GSH concentration in the TME, which facilitates desired drug release and optimizes the therapeutic efficiency [33,34,35].

## 2. Results

### 2.1. Characterization of Nanomaterials

To measure the synthesized nanomaterials, nuclear magnetic resonance (NMR) spectroscopy and the fourier transform infrared (FTIR) spectroscopy (Figure 1B) were used to detect the products in each step (Figure 1A). The FTIR of succinate acylated cholesterol (CHS) results showed two peaks of V_C=O_ at 1734 cm^−1^ and 1710 cm^−1^, which indicated the successful modification of the succinyl group. At the same time, the NMR spectrum showed that CHS had methylene peaks of succinyl at 2.46 ppm. The characteristic peak of mPssCOOH at 1700–1750 cm^−1^ on the infrared image is not obvious, which may be due to the use of mPEG with a molecular weight of 5000. Meanwhile, the hydrogen peaks of carboxyl group were approximately 8.1 ppm in NMR, which indicated the successful synthesis of mPssCOOH. In addition, 3,3′-dithiodipropionic acid derived a 2.9 ppm methylene peak, further proving its grafting on mPEG. For mPssLC, a similar FTIR peak as mPssCOOH at 2600–3200 cm^−1^ indicated successful methylene modification of PEG. At the same time, the 1–2 ppm of NMR peak corresponded to CHS, illustrating the grafting of CHS on mPssLC, and the inclusion peak at 8.2 ppm corresponded to the amino peak on polyethylene glycol. The above data indicated the successful synthesis of nanomaterials.

### 2.2. Characterization of Nanoparticles

For the in vitro experiments, we prepared the S2@PsPC1 NPs, which used the appropriate sorafinib and 2800Z concentration ratio, as previously reported [36]. For the in vivo experiments, we prepared the S2@PsPC2 nanoparticles that were loaded with the two drugs at a mass ratio of 1:1 to ensure that both drugs were at an effective concentration in vivo. We measured the hydrodynamics size and morphology of these NPs by TEM. The hydrodynamic dimensions of mPssPC NPs, S2@PsPC1 NPs, and S2@PsPC2 NPs were (300.3 ± 0.9) nm, (368.0 ± 5.9) nm, and (373.1 ± 11.6) nm, respectively (Figure 2A). The zeta potential of mPssPC, S2@PsPC1 NPs, and S2@PsPC2 NPs were (47.85 ± 0.98) mV, (50.45 ± 0.42) mV, and (53.2 ± 0.59) mV, respectively (Figure 2B). At the same time, the TEM image showed three NPs with a regular sphere (Figure 2C).

### 2.3. Drug-Loading Efficiency and Release Rates of S2@PsPCs NPs

We calculated the drug loading efficiency of NPs. The results showed that for S2@PsPC1 NPs, the drug-loading efficacy for sorafenib and 2800Z was 1.14 ± 0.06% and 17.6 ± 0.24%, respectively. For S2@PsPC2 NPs, it was 9.74 ± 0.12% and 9.65 ± 0.21%, respectively. We further performed an in vitro release experiment to detect the controlled drug release by using GSH to mimic the tumor microenvironment (TME) and pH 5.0 to mimic the acid environment in endosomes. As shown in Figure 3, both free sorafenib and 2800Z showed a high release rate by 8 h; the release rates for sorafenib were 74.44 ± 2.34% and 82.14 ± 1.36% (Figure 3A), and for 2800Z they are 66.25 ± 2.48% and 76.36 ± 1.01% (Figure 3B), respectively. In both cases, we observed enhanced drug release rates in the acidic conditions. In addition, we found that both NPs showed effective sustained drug release effects under TME conditions. For S2@PsPC1 NPs, the release rates of sorafenib and 2800Z by 48 h were 40.98 ± 1.29% and 36.73 ± 1.03%, and for S2@PsPC2 NPs, the release rates of sorafenib and 2800Z were 46.79 ± 1.83% and 33.20 ± 2.29%, respectively (Figure 3A,B). There was no significant difference in drug release between the two nanoparticles under normal conditions (Figure 3A,B). However, the drug release in the simulated tumor environment is much higher than in the normal environment (Figure 3A,B). For S2@mPsPC1 NPs, the release rate of sorafenib within 48 h raised to 73.63 ± 2.5% and 2800Z was 53.64 ± 2.44%. For S2@PsPC2 NPs, the release rate of sorafenib was 78.69 ± 1.73%, and 2800Z was 48.08 ± 2.21% (Figure 3A,B).

### 2.4. In Vitro Cell Uptake and the Distribution in Tumor Tissue

To measure cellular uptake of nanoparticles in vitro and in vivo, we prepared RED@PsPC NPs containing Cy5 and ICD@PsPC NPs containing indocyanine green (ICD) and measured the hydrodynamic dimensions; similar results were observed compared with other PsPC NPs (Appendix A). Huh7.5-luc cells were incubated with RED@PsPC NPs for 2 h and fluorescence were examined by flow cytometry. There was no fluorescent signal in untreated cells but fluorescent signals were detected as early as 1 h after treatment and further enhanced after 2 h of treatment (Figure 4A,B). The cellular uptake behaviors of RED@PsPC NPs were further investigated by fluorescence microscopy (Figure 4C). Similar to above shown results, the RED@PsPC NPs nanoparticles do have the ability to enter cells. We further evaluated the in vivo distribution of NPs by using ICD@PsPC NPs in tumor-bearing nude mice. As shown in Figure 4D, the NPs were equally diffused in vivo 1 h after the injection, and starting from 2 h after the injection, NPs gradually accumulation in the tumor region. By 6 h after the injection, NPs were predominantly concentrated in the tumor region.

### 2.5. Anticancer Effects of S2@PsPCs NPS In Vitro

To evaluate the cytotoxicity of sorafenib and 2800Z, Huh7.5-luc cells were treated with sorafenib, 2800Z, or in combination, and cell viabilities were detected by CCK8 assays (Figure 5A). The results shown that both sorafenib and 2800Z suppressed cell proliferation in a dose-dependent manner (Figure 5A). More importantly, we found that 2800Z significantly enhanced the toxicity of sorafenib, as previously reported [36]. We chose to prepare S2@PsPC1 NPs containing 80 μM 2800Z and 6 μM sorafenib, which showed minimal effects on cell proliferation when used alone, and tested the effects of drug-loaded NPs on liver cancer cells by CCK8 and colony formation assays (Figure 5B,C). Sorafenib@mPssPCs NPS (6 μM) showed no obvious effect on Huh 7.5-Luc cells, but S2@PsPCs NPS significantly decreased cell proliferation (Figure 5B). Similarly, S2@PsPCs significantly inhibited colony formation while the group of bank NPs or free Sor + 2800Z drug treatment showed no effects (Figure 5C). We further measured cell apoptosis by flow cytometry (Figure 5D) and the results indicated that S2@PsPCs were able to induce robust apoptosis of Huh 7.5-Luc cells compared with blank NPs and free Sor + 2800Z drug treatment (Figure 5E). This assay revealed that S2@PsPCs NPs treatment was able to efficiently inhibit Huh 7.5-Luc cell colony formation in comparison to free Sor + 2800Z drug treatment (Figure 5F).

### 2.6. Anticancer Effects of S2@PsPCs NPs In Vivo

We further explored the anti-tumor effects of S2@PsPCs NPs in vivo using a xenograft mouse model injected with Huh7.5 cells. Tumor bearing nude mice were treated with the free drugs or NPs for 14 days. We found that S2@PsPCs significantly inhibited tumor growth compare with blank NPs, while free drugs showed moderate effects (Figure 6A). We did not observe obvious body weight changes in either group (Figure 6B). After 14 days of treatment, we measured the tumor using bioluminescence imaging and the results showed that luminescence was not detectable in smaller tumors, which may be due to a detection limitation (Figure 6C). However, tumor growth in the S2@PsPCs NPs group was much slower and tumor size in the S2@PsPCs NPs group was significantly smaller than other groups by day 15 (Figure 6D). We performed ex vivo fluorescence examination to evaluate cancer cell metastasis in vivo and the results indicated that no metastasis occurred in the major organs (heart, liver, spleen, lungs, and kidneys) during the period of treatment (Figure 6E).

### 2.7. Evaluation of In Vivo Toxicity of S2@PsPCs NPS In Vivo

We evaluated in vivo toxicity of S2@PsPCs NPs. As shown in Figure 7, there are no abnormal changes in diet and behavior of all the treatment groups, suggesting that no serious side-effect was caused by the treatments of Sor-2800@mPssPCs NPs. We further examined pathological changes in different vital organs (lungs, heart, spleen, liver, and kidneys) by performing the H&E staining. The results indicated that there were no pathological changes observed in the lungs, spleen, kidneys, or heart in all treatment groups.

### 2.8. Immuno-Histopathology Analysis within the Tumor after S2@PsPCs NPS Treatment

We finally performed immuno-histochemistry analysis to test the apoptosis and proliferation levels of the tumor. As shown in Figure 8A, only few PCNA positive cells were observed in the tumor tissues in the S2@PsPCs NPs group in comparison with other groups. To further evaluate whether S2@PsPCs NPs could promote cell apoptosis in tumor tissues, we evaluated the protein expression of cleaved-caspase 3 (Figure 8B). The expression of the cleaved-caspase 3 level in the S2@PsPCs NPs group was much higher than in other groups, suggesting that the S2@PsPCs NPs could inhibit proliferation and promote apoptosis in tumor cells.

## 3. Discussion

In this study, we developed a dual drug loading system by using polylysine as the nanomaterial modified with cholesterol and GSH-sensitive PEG (mPssPC) to enable the formation of polymer nanoparticles with a hydrophobic core and hydrophilic surface. Sorafenib and SIRT7 inhibitor 2800Z dual-loaded NPs were designed, and we evaluated the efficacy for the treatment of liver cancer. The average sizes and zeta potentials of S2@mPssPC1 NPs and S2@PsPC2 NPs were (368.0 ± 5.9) nm and (50.45 ± 0.42) mV, and (373.1 ± 11.6) nm and (53.2 ± 0.59) mV, respectively. The release of the drugs from these nanoparticles was significantly accelerated in the TME and acid simulating conditions, which exhibited GSH and pH sensitive drug release behavior. S2@PsPC1 NPs inhibited proliferation and induced apoptosis of liver cancer cells in vitro. The in vivo study further revealed that S2@PsPC2 NPs showed high specificity of targeting the liver cancer tissue but low affinity and toxicity to main organs including the heart, kidneys, lungs, and liver. Our data thus provides a new platform for a co-delivery system with high affinity to liver tumors and further approved the therapeutic effects of combinational therapy with a SIRT7 inhibitor and sorafenib, which offers a new therapeutic option in treatment of human liver cancer.

SIRT7 is a NAD^+^-dependent histone deacetylase that plays critical roles in human cancer by modulating key processes linked to cell fate determination and oncogenesis such as genome stability, DNA damage repair, and apoptosis [37,38]. Altered SIRT7 expression is frequently observed in human cancers and high SIRT7 is associated with an aggressive cancer phenotype and poor clinical outcomes [39,40,41]. Inactivate SIRT7 impairs cancer transformation, increases chemosensitivity, and reverses metastatic phenotypes in cancer [9,42]. By using virtual screening, we have previously shown that 2800Z specifically inhibits SIRT7 enzyme activity and exhibits synergistic effects to sorafenib, mainly through the SIRT7/p53/NOXA axis [15,16]. In this study, we further demonstrated that NPs with two drugs showed high specificity of targeting the tumor but low affinity and toxicity to other organs. Mechanisms underlying mPssPC2 NPs mediated inhibition of liver cancer are not fully understood, and whether SIRT7 is responsible for this inhibition is currently under investigation in the lab.

Because of the EPR effect of the tumor microenvironment for small-sized nanoparticles, the synthesized mPssPCs NPs exhibited significant retention at the tumor site [27], and the surface of NPs can be specifically modified to achieve TME responsiveness [42]. In present study, we utilized GSH-responsive disulfide bonds linking polyethylene glycol and polylysine for dual loading of 2800Z with sorafenib to significantly enhance the cellular uptake of mPssPCs NPs by liver cancer cells. Our data indicated that the use of GSH-responsive disulfide bonds enabled nanoparticles to achieve enrichment at the tumor site. In addition, we observed similar tumor targeting abilities in other malignancies by using the same system [43,44]. Of note, the presence of polyethylene glycol ensures that S2@mPsPC1 NPs were not cleared by the immune system, and the surface-modified polyethylene glycol increased the in vivo circulation time of S2@mPsPC NPs [45,46], which enable their targeted action. Moreover, researchers have modified polyethylene glycol on the surface of polylysine to improve its biocompatibility, which can be used as a polymeric cationic material to allow endosomal escape through the proton sponge effect after uptake by cells [47]. A new way of polyethylene glycol-modified poly(lysine) nanoparticles loaded ubenimex for hepatocellular carcinoma therapy has shown promising results in experiments [48].

We showed that the synthesized S2Z@mPsPC2 NPs exhibited high specificity of targeting the tumor in mice; however, the efficiency of achieving targeting based on the EPR effect in humans requires further investigation. Due to the high heterogeneity of human liver cancer, EPR effects are highly variable in patients [27]. In fact, even in the same patients, EPR effects are also influenced by the TME, including the degree of angiogenesis, perivascular tumor growth, and physicochemical properties such as size, charge, and surface modification of the nanodrug [49]. Although active targeting based on the target-ligand binding reaction is more efficient than passive targeting based on EPR, the complexity and heterogeneity of tumors themselves make it difficult to find antigens specifically expressed only on the surface of tumor cells and the tumor cells with different genotypes and phenotypes cannot be targeted by the same ligands [49,50].

Although the safety of S2@mPsPC NPs was tested in this study through cellular and animal experiments, there are significant differences between human and mouse bodies. We have previously showed that the micelles formed by nanoparticles are prone to aggregation under in vivo physiological conditions due to their instability, which can lead to side effects such as embolism and alteration of biodistribution *in vivo*, which seriously affects their safety in vivo and is a major limitation for the further clinical use of nanoparticles [51]. Therefore, the safety of our synthesized S2@mPsPC NPs still deserves further investigations. In particular, the investigation on the interaction of polylysine nanoparticles and biological organisms is essential to fully explore the prospects of polylysine nanomedicines for medical applications.

## 4. Materials and Methods

### 4.1. Cells and Animals

Huh7.5 cells were purchased from Procell (Wuhan, China) and Huh7.5-luc cells were generated by stable transfecting firefly luciferase plasmid DNA (pGL-3 promoter vector) in Huh7.5 cells. All cells were grown in Dulbecco’s modified Eagle’s medium (Gibco, New York, NY, USA) containing 10% fetal bovine serum (Gibco, New York, NY, USA) and 1% penicillin/streptomycin (Thermo fisher, Waltham, MA, USA) at 37 °C in a humidified 5% CO_2_ incubator.

BALB/c nude mice aged 3 weeks were purchased from Gempharmatech (Nanjing, China). Mice were housed in a temperature-controlled (relative humidity of 50–60% and controlled room temperature of 20–22 °C), pathogen-free environment with 12 h light–dark cycles. All mice were treated with standard laboratory feed and water. All animal handling procedures were approved by the Institutional Animal Care and Use Committee at Hunan Normal University School of Medicine (Protocol 2020020).

### 4.2. Antibodies and Chemicals

1-Ethyl-3-(3-dimethylaminopropyl) carbodiimide hydrochloride (EDCI), 4-Dimethylaminopyridine (DMAP), and mPEG5000 were purchased from Aladdin (Shanghai, China). Dimethyl sulfoxide (DMSO), Succinic anhydride (SA), and cholesterol were purchased from Sinopharm Chemical Reagent Co., Ltd. (Shanghai, China). 3,3′-Dithiodipropionic acid and Poly-(L-lysine) (PLLS) were from Shanghai Macklin Biochemical Co., Ltd. (Shanghai, China). Sorafenib (BAY 43–9006) was purchased from Selleck Chemicals (Shanghai, China), and 2800Z was purchased from Chemdiv (San Diego, CA, USA). PARP (9532) and cleaved Caspase-3 (9662) were purchased from Cell Signaling Technology (Danvers, MA, USA). PCNA (ab29) was purchased from Abcam (Cambridge, MA, USA).

### 4.3. Synthesis of Succinate Acylated Cholesterol (CHS)

Two grams of cholesterol and 1 g of succinic anhydride was dissolved in pyridine and mixed on the magnetic stirrer (DF-101S, Shanghai Licheng-BX Instrument Technology co., Ltd. (Shanghai, China)) at 50 °C. After 2 days, the reaction mixture was poured into 4 °C water. Hydrochloric acid was used to adjust the pH to 3.0. Then, the precipitation was separated by filtration, lyophilized, and dissolved in 20 mL of ethanol ethyl acetate mixed solution (1:1 of volume ratio) at 60 °C. Finally, the mixed solution was placed at −20 °C for recrystallization, and the product was filtered and dried to obtain CHS.

### 4.4. Synthesis of mPssCOOH

Two grams of 3,3′-Dithiodipropionic acid and 0.6 g of DMAP were dissolved in DMSO and mixed on the magnetic stirrer 40 °C for 30 min. Next, 5 g of mPEG5000 was added to the mixture. After 2 days, a dialysis bag (3000 Da) was used to dialyze the reaction mixture in 2000 mL deionized water, and fresh deionized water was added to replace the obsolete water per hour. The dialysis solution was lyophilized to obtain the mPssCOOH product.

### 4.5. Synthesis of mPssPC

Eight-hundred milligrams of mPssCOOH, 0.5 g CHS, 0.2 g DMAP, and 0.4 g EDCI were dissolved in DMSO and mixed on the magnetic stirrer at 40 °C for 30 min. Next, 1 g of PLLS was added to the mixture. After 2 days, a dialysis bag (7000–14,000 Da) was used to dialyze the reaction mixture in 2000 mL deionized water, and fresh deionized water was added to replace the obsolete water per hour. The dialysis solution was lyophilized to obtain the mPssPC product.

### 4.6. FTIR and NMR of Nanomaterials

Two milligrams of CHS, mPssCOOH, and mPssPC were mixed with potassium bromide solid, and grinded. Then, the mixed powders were pressed on the tablet machine, and scanned in the spectrum at 4000–400 cm^−1^ on the FTIR instrument (Resolution: 1, iS20, Themofisher). Ten milligrams of CHS, mPssCOOH, and mPssPC were dissolved in 1 mL of (Methyl sulfoxide)-d6 (DMSO-d6) and put into the NMR sample tubes. 1H NMR spectrum was performed by scanning the absorption from −3 ppm to 15 ppm (500 MHz, AV-500, Bruker, Billerica, MA, USA).

### 4.7. Preparation of PsPC NPs and S2@PsPCs NPs

Five milligrams of mPssPC was dissolved in 5 mL of DMSO. Then, a dialysis bag (7000–14,000 Da) was used to dialyze the mixed solution in PBS. Fresh PBS was added to replace the obsolete PBS per hour. Then, 180 μg of sorafenib, 2.8 mg of 2800Z, and 12 mg of mPssPC were added to prepare S2@PsPC1 NPs. One milligram of sorafenib, 1 mg of 2800Z, and 8 mg of mPssPC were added to prepare S2@PsPC2 NPs. Two milligrams of indocyanine green (ICG) and 8 mg of mPssPC were added to prepare ICD@PsPC NPs. We then added 1 mg of Cy5 and 4 mg of mPssPC to prepare RED@PsPC NPs.

### 4.8. Size and Morphology Measurement of Nanoparticles

Dimensions and the zeta potential of NPs in PCS8501 or DTS1070 were measured with default settings by dynamic light scattering (DLS, Zetasizer Nano ZS90, Malvern). The test conditions were argon ion laser, wavelength 658 nm, temperature 25 ± 0.1 °C, and DLS angle 90°. Zeta-potential was determined at the same time. The operating conditions were 11.4 v cm^−1^, 13.0 mA, and 25 °C. For morphological measurement, NPs were dripped onto the copper mesh (200 mesh size), dried, and dyed with 1.5% phosphotungstic acid solution. The size and morphology were observed by transmission electron microscope (200 kV, TEM, TecnaiG2 F20, FEI).

### 4.9. Evaluation of Drug-Loading Efficiency and Drug Release

Sorafenib and 2800Z were dissolved in DMSO and prepared at concentrations of 0, 2, 4, 6, 8, and 10 μg/mL; absorbance was detected at 214 nm and 265 nm (UV-1900i, SHIMADZU). To determine the drug-loading of the S2@PsPCs NPs, the prepared NPs were diluted 10 times in DMSO, after further diluted 10 times in water, and the absorbance measured at 214 nm and 265 nm with control blank NPs. To measure the drug release of S2@PsPCs NPs, 5 mL of above NPs were placed separately in the dialysis bag, dialyzed in 50 mL of PBS (pH = 7.4 or 5.0 with GSH), and placed on a shaker (37 °C, 75 rpm). The PBS was taken out from the dialysis bag at various time points, the solution was diluted 10 times with DMSO, and the content of the released drug was calculated through absorbance at 214 nm and 265 nm. The volume of PBS at the time point was represent as Vt and the drug concentration was represented as Ct, and the release rate was calculated according to the following formula:(1)$$Q\%=\left(V0\times t0+\overset{48}{\underset{t=0}{\sum }}\mathrm{Vt}\times \mathrm{Ct}\;\right)/m_{\mathrm{drug}}\times 100\%$$
where m_drug_ is the total mass of single drug in nanoparticles in dialysis bag, t is the time point for replacing PBS (t = 1, 2, 4, 8, 16, 24, and 48 h; both V_0_ and C_0_ are equal to zero).

### 4.10. Cellular Uptake Assessment In Vitro

The cellular uptake of the RED@PsPC NPs were evaluated by measuring the fluorescence properties. In brief, Huh7.5-luc cells were seeded in 6-well plates at 10^6^ cells per well for 24 h. The cells were then incubated with 2 mL of DMEM containing RED@PsPC NPs for 1 h or 2 h. After washing with PBS (0.01 M, pH7.4) (Gibco, New York, NY, USA) twice, cells were collected by centrifuging at 1000 rpm for 5 min and the fluorescence signals were observed using a fluorescence microscopy (Leica, Wetzlar, Germany) and measured by flow cytometry (BD Biosciences, Becton, NJ, USA).

### 4.11. Distribution of Nanoparticles In Vivo

When the tumor volume reached ~400 mm^3^, mice were given a tail vein injection of ICD@PsPC NPs. The fluorescence intensities were photographically recorded by IVIS Lumina LT (PerkinElmer, Waltham, MA, USA) at 0, 0.5, 1, 2, 4, and 6 h after injection.

### 4.12. Cell Counting Kit-8 (CCK8) Assay

The CCK8 assay was performed as previously described [16]. Huh7.5-luc cells were plated onto 96-well microtiter plates (Corning, Tewksbury, MA, USA) at a density of 3000 cells per well for overnight. Cells were then treated with NPs, Sorafenib, 2800Z, or S2@PsPC1 NPs for additional 24 h. Then the culture medium was changed with 10% CCK-8 solution (Dojin Laboratories, Kumamoto, Japan) and incubated for 2 h at 37 °C. Then, absorbance was determined at 450 nm using microplate reader.

### 4.13. Apoptosis Assay

The apoptosis of Huh7.5-luc cells was examined by the allophycocyanin (APC) annexin V and propidium iodide (PI) apoptosis detection kit (Yeasen, Shanghai, China). Huh7.5-luc cells were seeded onto six-well plates (Corning, Tewksbury, MA, USA) at a density of 1 × 10^6^ cells/well, and after treating with NPs, Sorafenib and 2800Z, or S2@PsPCs NPs for 24 h, they were collected by centrifuging at 1000 rpm for 5 min. Meanwhile, the cells were labeled by the non-vital dye PI and annexin V-fluorescein isothiocyanate (FITC), which was employed to detect different cell populations, including the intact cells (i.e., FITC^–^PI^–^ cells), the early apoptotic cells (i.e., FITC^+^PI^–^ cells), and the late apoptotic cells (i.e., FITC^+^PI^+^ cells) using flow cytometry (BD Biosciences, Becton, NJ, USA).

### 4.14. Colony Formation Assay

Huh7.5-luc cells were added to 24-well plates at a density of 3 × 10^3^ cells per well and then incubated for 7–10 additional days in the presence of NPs, Sorafenib and 2800Z, or S2@PsPCs NPs. Cells were fixed by 10% formaldehyde and then stained with 0.1% crystal violet. Colony numbers were counted by at least five random fields.

### 4.15. Murine Xenograft Models

BALB/c nude mice were subcutaneously implanted in the right flank with 5 × 10^6^ Huh7.5-luc cells. When tumors had grown to 40–60 mm^3^ in size, mice were randomly divided into control (100 µL PBS), bland-NPs, free drug combination containing 2800Z (1 mg/kg/day), sorafenib (1 mg/kg/day), and S2@PsPCs NPs loaded with 2800Z (1 mg/kg/day) and sorafenib (1 mg/kg/day) groups (*n* = 5 each group). All treatments were administered via tail vein once every 2 days for 2 weeks. Tumor volumes and body weight were measured every 2 days, and tumor volume was calculated as follows: volume = 1/2 (length × width^2^). All mice were euthanized with sodium pentobarbital (Sigma Aldrich, St Louis, MO, USA) injection (150 mg/kg) according to AVMA Guidelines for the Euthanasia of Animals, and tumors were removed for further analysis.

### 4.16. In Vivo Bioluminescence Imaging

In vivo imaging was performed on tumor bearing nude mice at day 14. Prior to imaging, the luciferin aqueous solution (150 mg·kg^−1^) was injected via tail vein injection and followed by a 5 min waiting period. Mice examined for quantification of Huh7.5-luc tumor growth were imaged from the lateral decubitus position. The tumor bioluminescence was observed by a IVIS Lumina LT in vivo imaging system (PerkinElmer, Waltham, MA, USA) and normalized against the initial flux. The mice were sacrificed later, and the heart, liver, spleen, lungs, kidneys, and tumor were collected immediately. The fluorescence intensities in the different tissues were photographically recorded via IVIS Lumina LT system.

### 4.17. Histological Analysis

Animals were euthanized and organs including the heart, liver, spleen, lungs, kidneys, and tumor were collected and fixed in 4% neutral-buffered formalin to prepare histologic slides. Samples of organs were then stained with hematoxylin and eosin (H&E). Immunohistochemistry was performed as previously described [16]. After deparaffinization and rehydration, antigen retrieval was achieved by heating in a pressure cooker for 5 min in 10 mM of sodium citrate (pH 6). Sections were rinsed three time in PBS/PBS-T (0.1% Tween-20) and incubated in 4% FBS to block for 60 min. After removal of blocking solution, slides were placed into a humidified chamber and incubated overnight at 4 °C with primary antibodies in blocking buffer (4% normal goat serum in PBS). After washing with PBS, slides were covered with SignalStain Boost IHC Detection Reagent (Cell Signaling Technologies, Boston, MA, USA) for 30 min at room temperature. After washing two times with PBS-T, the DAB color developer (Biosharp, Hefei, China) was applied, and the slides were incubated for 5–10 min and counterstained with hematoxylin. Images were acquired using a Zeiss Axiolab 5 Digital Lab Microscope (Carl Zeiss AG, Jena, Germany).

### 4.18. Statistical Analysis

Data are presented as mean ± SEM. Statistical analysis were performed by using GraphPad Prism 6. Statistical significance between groups was calculated by using one-way ANOVA followed by Turkey’s test. Statistical significance between two groups was calculated by the two-tailed unpaired Student’s t-test. Variance between groups met the assumptions of the appropriate test. Unless otherwise stated, a *p*-value of <0.05 was considered statistically significant.

## 5. Conclusions

S2@PsPCs NPs co-delivery system with small hydrodynamic dimensions and a positive surface charge was successfully designed. The release of drug exhibited pH sensitivity and was significantly accelerated in an acid release medium simulating the tumor environment. In addition, S2@PsPC NPs inhibited the proliferation and induced apoptosis of liver cancer cells in vitro. The in vivo study further revealed that S2@PsPCs showed high specificity to the liver cancer but low affinity and toxicity to main organs including the heart, kidneys, lungs, and liver. Our data thus provides a new platform for a co-delivery system with high affinity to liver tumors and a new therapeutic option in treatment of human liver cancer.

## Figures and Tables

**Figure 1 pharmaceuticals-16-00119-f001:**
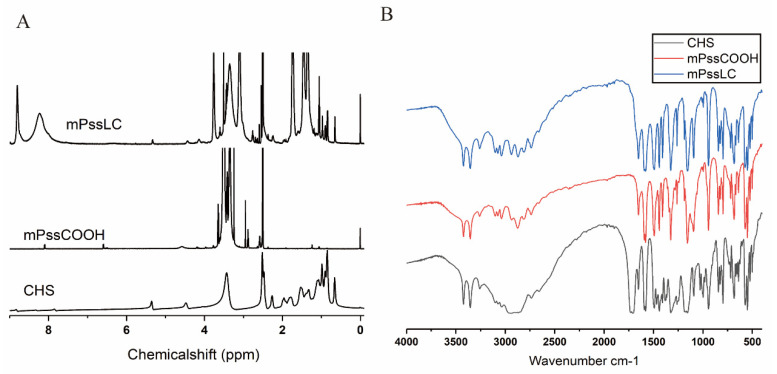
Characterization of CHS, mPssCOOH, and mPssLC. The synthesis of the material was verified by 1H NMR (**A**) and FTIR (**B**).

**Figure 2 pharmaceuticals-16-00119-f002:**
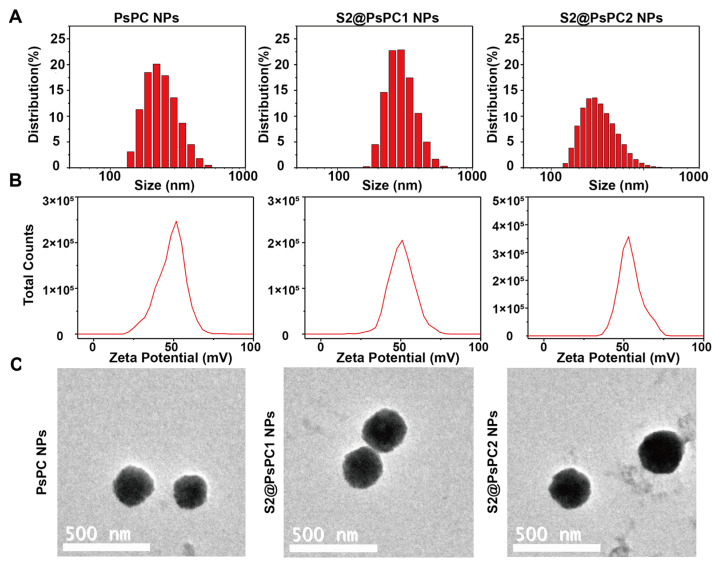
Characterization of PsPC NPs (Blank NPs), S2@PsPC1 NPs, and S2@PsPC2 NPs. The size distribution (**A**) and zeta potential (**B**) were detected by dynamic light scattering (DLS). (**C**) The size of NPs were detected by transmission electron microscopy (TEM). Scale bar indicates 500 nm.

**Figure 3 pharmaceuticals-16-00119-f003:**
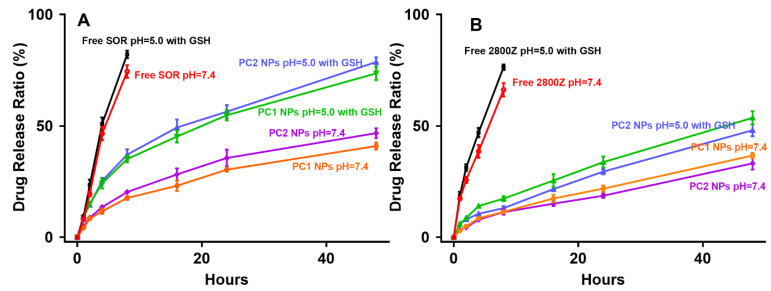
Drug release rates of sorafenib (SOR) and 2800Z from free drug, S2@PsPC1 NPs (PC1), and S2@PsPC2 NPs (PC2). SOR (**A**) and 2800Z (**B**) release of free drugs, PC1, and PC2 under physiological conditions (pH = 7.4) or a simulated tumor-endosome microenvironment (pH = 5.0 with 10 mM of GSH) during 48 h.

**Figure 4 pharmaceuticals-16-00119-f004:**
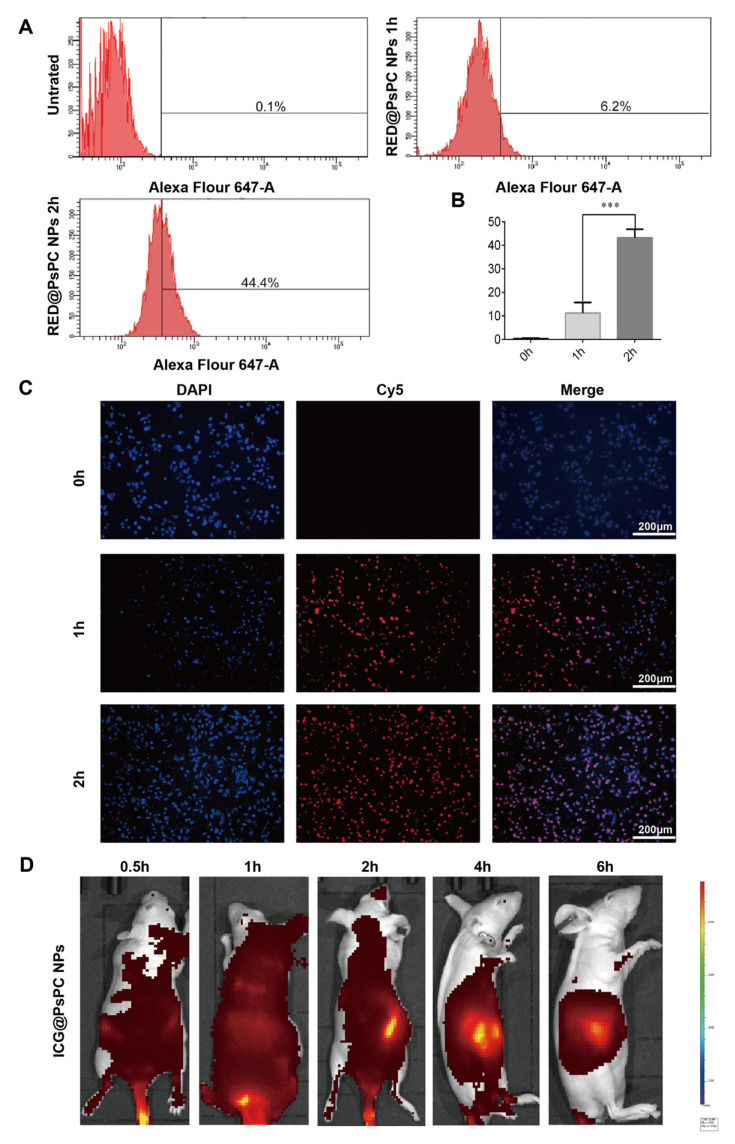
The cellular uptake of nanoparticles in vitro and its tumor-target ability in vivo. Huh7.5-luc cells were incubated with RED@PsPC NPs containing Cy5 for various time points, as indicated. Cells were washed by PBS and red stained cells were measured by flow cytometry (**A**), with data being quantified in (**B**), *** *p* < 0.001, *n* = 3. (**C**) Cells in A were evaluated by microscopy. Scale bar indicates 200 µm. (**D**) Tumor bearing mice were received tail vein injection of ICD@PsPC NPs containing indocyanine green (ICG). The fluorescence intensities were photographically recorded by IVIS Lumina LT at various time points, as indicated.

**Figure 5 pharmaceuticals-16-00119-f005:**
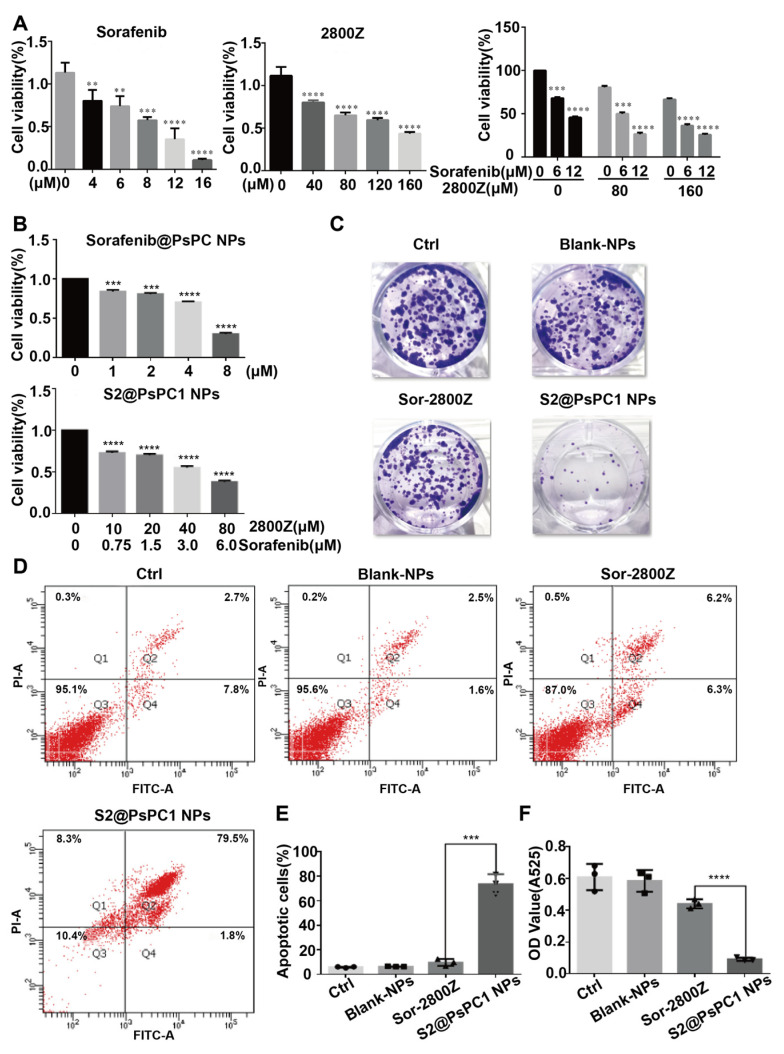
The S2@PsPC1 NPs inhibit cell proliferation, colony-forming activity, and promote cell apoptosis of Huh7.5-luc cells. (**A**) Huh7.5-luc cells were treated with a range of concentrations of sorafenib, 2800Z, or a combination, and cell viability was assessed 24 h after treatment using the CCK8 assay. (**B**) The viability of Huh7.5-luc cells was assessed at 24 h post Sorafenib@mPssPCs NPs or S2@PsPCs NPs treatment. (**C**) Huh7.5-luc cells was untreated or treated with free drug combinations of sorafenib and S2@mPssPC1 NPs for 10 days, with data being quantified in (**F**). (**D**) Representative flow cytometry analysis of cells stained with propidium iodide (*y*-axis) and annexin V-FITC (*x*-axis), with data being quantified in (**E**). Graphs show mean ± SEM of at least three independent experiments (** *p* < 0.01, *** *p* < 0.001, **** *p* < 0.0001, *n* = 3).

**Figure 6 pharmaceuticals-16-00119-f006:**
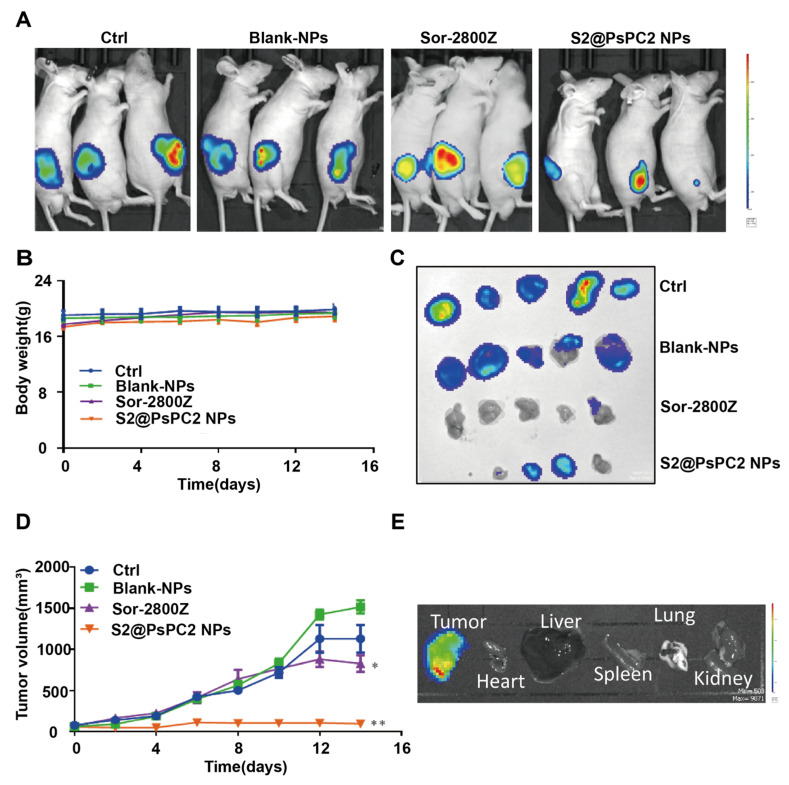
In vivo assessment of anticancer effects of S2@PsPCs NPs. (**A**) Huh7.5-luc cell were implanted in nude mice. When the tumor volume reached ~400 mm^3^, mice were treated with control (PBS), blank NPs, free sorafenib and 2800Z in combination, or S2@PsPC2 NPs for 14 days. Bioluminescence images of mice at day 15 in each treatment group. (**B**) Murine body weight during the period of treatment. (**C**) Bioluminescence images of the tumor. (**D**) Tumor volume changes during the period of treatment. (**E**) The ex vivo optical images of tumors and vital organs of Huh7.5-luc tumor-bearing mice (* *p* < 0.05, ** *p* < 0.01 vs Blank NPs, *n* = 3).

**Figure 7 pharmaceuticals-16-00119-f007:**
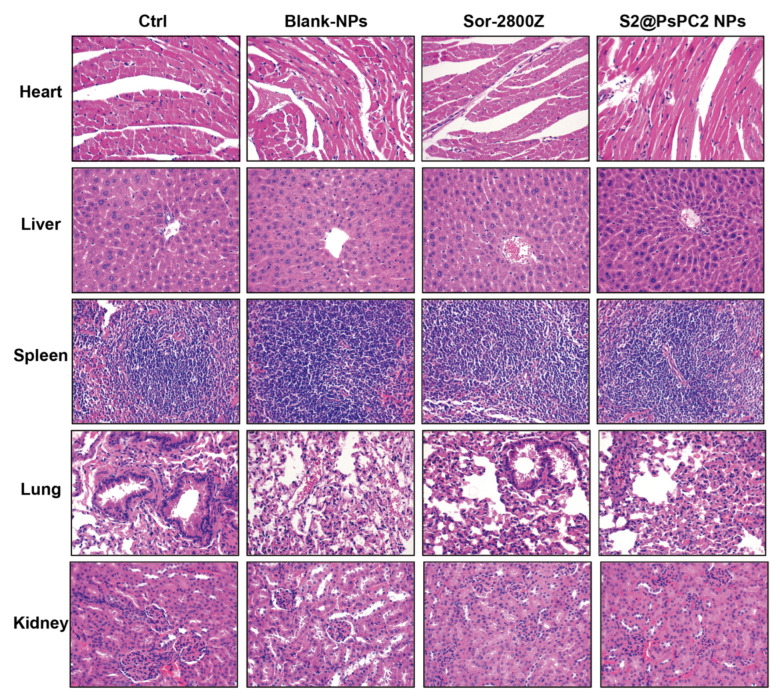
Histological assessment of organs with H&E staining in mice treated with control (PBS), blank NPs, free sorafenib and 2800Z combination (Sor-2800Z), or S2@PsPC2 NPs for 14 days. Images showed representative sections from five mice of each group.

**Figure 8 pharmaceuticals-16-00119-f008:**
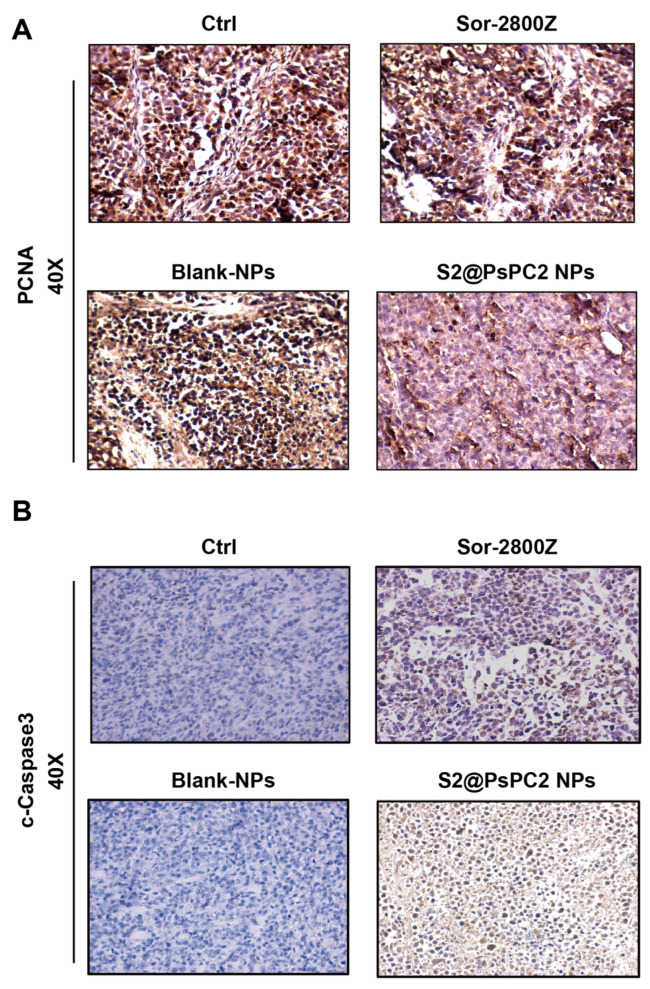
Representative IHC staining of PCNA (**A**) and cleaved-caspase3 (C-caspase) (**B**) in tumors of mice treated with control (PBS, Ctrl), blank NPs, free sorafenib and 2800Z combination (Sor-2800Z), or S2@PsPC2 NPs for 14 days. Images showed representative sections from five mice of each group.

## Data Availability

The authors declare that all data supporting the findings of this study are available within the paper.

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
