# Peer review of "Sorafenib/2800Z Co-Loaded into Cholesterol and PEG Grafted Polylysine NPs for Liver Cancer Treatment"

_pharmaceuticals, 2023, doi:10.3390/ph16010119_

Round 1
Reviewer 1 Report
The authors try to present an interesting continuation of the previous study. Several
- 1. Row 86: Huh7.5- luc cells were purchased from Procell (Wuhan, China) or prepared? Please, write the correct one. Describe the preparation of luciferase plasmid DNA (pGL-3 promoter vector).
- 2. Please specify the conditions and solutions used for dialysis in all cases where dialysis was performed.
- 3. Please, indicate the DLS default settings
- 4. In all cases (DLS, TEM, FTIR), add the description of the used equipment and the analysis conditions performed.
- 5. Rows 142-153: Please, describe the dilutions, the choice of the wavelength used, and the equipment.
- 6. The preparation of RED@mPssPC and ICG@mPssPC NPs is not described somewhere. Please, describe, or add the correct citation, or show the characterization of both NPs. Please, write it in detail.
- 7. Row 172: CCK-8 -It is a kit for colorimetric assays to determine cell viability in cell proliferation and cytotoxicity assays. Please, describe the assay adequately.
- 8. The Figures within the text are not numbered and the reviewers have to guess which legend belongs to each figure. Please, correct it!
- 9. Please describe why Huh7.5-luc cells were used in FACS analysis instead of Huh7.5 cells. Since the luciferase assay was not used here.
- 10. Row 213: The section “2.17. Histological Analysis” is actually “Immunohistochemical staining” and has to be written correctly. Any description of the antibody used is missing. Please, add.
- 11. In the section “Results”, the figure between rows 249-250 (assuming Figure 1) shows results from NMR. This analysis is not described, even such section exists “2.6. FTIR and NMR of nanomaterials”. Please, describe the NMR experiments.
- 12. Please describe the difference between SOR-2800Z@mPssPC1 and SOR-2800Z@mPssPC2.
etc...
Author Response
- Row 86: Huh7.5- luc cells were purchased from Procell (Wuhan, China) or prepared? Please, write the correct one. Describe the preparation of luciferase plasmid DNA (pGL-3 promoter vector).
Thank you for pointing out this error. We purchased Huh7.5 cells from Procell (Wuhan, China) and stably transfected luciferase plasmid DNA (pGL-3 promoter vector) in Huh7.5 cells to make Huh7.5-Luc cells which used in this study. We have made the correction in revised version.
- Please specify the conditions and solutions used for dialysis in all cases where dialysis was performed.
Thank you for this important suggestion. The dialysis was performed by using dialysis bag (7000-14000 Da) which was placed in 2000 mL deionized water and the obsolete water were replaced by fresh deionized water every hour. We have specified these in section 4, row 1034-1035, 1075-1076, 141-142.
- Please, indicate the DLS default settings
We have added the DLS default settings in section 4.8, raw 1072-1079.
- In all cases (DLS TEM, FTIR), add the description of the used equipment and the analysis conditions performed.
We have added description of equipment and analysis settings in section 4.6 and 4.8, raw 1048-1062 and 1072-1079.
- Rows 142-153: Please, describe the dilutions, the choice of the wavelength used, and the equipment.
We have added these in section 4.9, raw 1083-1084.
- The preparation of RED@mPssPC and ICG@mPssPC NPs is not described somewhere. Please, describe, or add the correct citation, or show the characterization of both NPs. Please, write it in detail.
We have added the preparation of RED@mPssPC and ICG@mPssPC NPs to the section 4.7, raw 1068-1070. Compared with other nanoparticles, the RED@mPssPC and ICG@mPssPC NPs we prepared were different only in different drugs loaded. Therefore, we just tested the hydrodynamic dimension, and we added the data to the supplement information and mention this in text.
- Row 172: CCK-8 -It is a kit for colorimetric assays to determine cell viability in cell proliferation and cytotoxicity assays. Please, describe the assay adequately.
The CCK8 assay was performed as previously and we described the assay more detailly in our revised version from raw 1050-1055.
- The Figures within the text are not numbered and the reviewers have to guess which legend belongs to each figure. Please, correct it!
Thank you for pointing out this error, we have made corrections in the revised version.
- Please describe why Huh7.5-luc cells were used in FACS analysis instead of Huh7.5 cells. Since the luciferase assay was not used here.
We used Huh7.5-Luc cell for in vivo experiments, and for in vitro testing, we using this cells in order to make consistence experimental settings.
- Row 213: The section “2.17. Histological Analysis” is actually “Immunohistochemical staining” and has to be written correctly. Any description of the antibody used is missing. Please, add.
Thank you for point out this error, we added the description of antibodies used for IHC staining. However, this section including H&E staining which belongs to histological analysis thus we retained the section name.
- In the section “Results”, the figure between rows 249-250 (assuming Figure 1) shows results from NMR. This analysis is not described, even such section exists “2.6. FTIR and NMR of nanomaterials”. Please, describe the NMR experiments.
Thanks for your suggestion. We have added more describe of the NMR experiments in section 2.1, raw 506-515 to illustrate materials synthesis.
- Please describe the difference between SOR-2800Z@mPssPC1 and SOR-2800Z@mPssPC2.
Thanks for your suggestion. Because of the significant difference in IC50 between SOR and 2800Z, we prepared the SOR-2800Z@mPssPC1 NPs for experiments in vitro according to synergistic effects of two drugs. However, to ensure both drugs were at an effective concentration in vivo, we prepared SOR-2800Z@mPssPC2 NPs which had the 1:1 ratio of SOR to 2800Z. We have added description of these differences in section 4.2, raw 512-515 of the revised manuscript.
Reviewer 2 Report
The authors have developed a novel nanocarrier system for combining 2800Z, a new SIRT7 inhibitor, with sorafenib. The study is novel and of high interest to the readers of Pharmaceutics, and therefore I recommend publication after the following minor corrections:
Minor comments:
1. I recommend using 'approximately' rather than 'around', specifically within the abstract.
2. The use of SOR-2800Z@mPssPC as an abbreviation is very complex and difficult to follow. I recommend using simpler abbreviations throughout.
3. The rationale for delivering sorafenib and 2800Z within a nanocarrier system could be better highlighted within the abstract. While the rationale can be assumed, it may not be obvious for all readers.
4. The rationale for selecting PLLS grafted cholesterol and GSH sensitive PEG groups for the nanoparticles could be more clearly presented within the final paragraph of the introduction.
5. The authors report drug loading efficacy in section 3.3. What is this? Should this be drug loading efficiency? Is this based on equation 1?
6. What buffer was used for the pH 5 drug release studies? Why did the authors not use Artificial Lysosomal Fluid to replicate the lysosomal environment?
7. Figures are missing captions!! This must be corrected before publication. It is impossible to understand what is shown in each figure in the current manuscript.
8. Figure 4C is difficult to interpret in the current form. What are the scale bars? The brightness is also very low and needs to be increased to see the micrographs.
Author Response
- I recommend using 'approximately' rather than 'around', specifically within the abstract.
Thank you for the suggestion, we made the corrections throughout the manuscript in revised version.
- The use of SOR-2800Z@mPssPC as an abbreviation is very complex and difficult to follow. I recommend using simpler abbreviations throughout.
Thank you for this important suggestion. We have simplified the abbreviation to S2@PsPC1 for in vitro and S2@PsPC2 for in vivo experiment in the revised version.
- The rationale for delivering sorafenib and 2800Z within a nanocarrier system could be better highlighted within the abstract. While the rationale can be assumed, it may not be obvious for all readers.
Thank you for this important suggestion. We have modified our abstract to better address rational of our study and dual loading nanoparticles.
- The rationale for selecting PLLS grafted cholesterol and GSH sensitive PEG groups for the nanoparticles could be more clearly presented within the final paragraph of the introduction.
Thanks for your suggestion. We have added the rationale within the final paragraph of the introduction, raw 132-138.
- The authors report drug loading efficacy in section 3.3. What is this? Should this be drug loading efficiency?
Thanks for your pointing out this error. It should be drug loading efficiency and we made correction in revised version.
- What buffer was used for the pH 5 drug release studies? Why did the authors not use Artificial Lysosomal Fluid to replicate the lysosomal environment?
Thanks for your suggestions. HCl-PBS solution was used for the pH 5.0 drug release studies. The PEG protects nanoparticles from being engulfed by immune cells, but at the same time reduces tumor cells uptake. The PEG grafted disulfide bond has bond-breaking in the TEM with high GSH and then the uncoated NPs were easier to be ingested. Because it is possible that the disulfide bond broken before entering endosome, we thus did not used the Artificial Lysosomal Fluid.
- Figures are missing captions!! This must be corrected before publication. It is impossible to understand what is shown in each figure in the current manuscript.
Thank you for pointing out this error, we made the corrections in revised version.
- Figure 4C is difficult to interpret in the current form. What are the scale bars? The brightness is also very low and needs to be increased to see the micrographs.
Thank you for pointing out this error. We added scale bar and increased brightness of images in revised version.